IFT-UAM/CSIC-24-076
May 27th, 2024

# Gravitational higher–form symmetries
# and the origin of hidden symmetries
# in Kaluza–Klein compactifications

*Carmen Gómez-Fayrén,*[a] *Tomás Ortín*[b] *and Matteo Zatti*[c]

*Instituto de Física Teórica UAM/CSIC*

*C/ Nicolás Cabrera, 13–15, C.U. Cantoblanco, E-28049 Madrid, Spain*

## Abstract

We show that, in presence of isometries and non-trivial topology, the Einstein–Hilbert action is invariant under certain transformations of the metric which are not diffeomorphisms. These transformations are similar to the higher-form symmetries of field theories with $p$-form fields. In the context of toroidal Kaluza–Klein compactifications, we show that these symmetries give rise to some of the "hidden symmetries" (dualities) of the dimensionally-reduced theories.

[a]Email: carmen.gomez-fayren[at]estudiante.uam.es
[b]Email: tomas.ortin[at]csic.es
[c]Email: matteo.zatti[at]estudiante.uam.es

# 1  Introduction

The main theme of Kaluza–Klein (KK) theories[1] is that the symmetries of the lower-dimensional theory obtained by compactification have a higher-dimensional purely gravitational (that is, geometric) origin. It is not difficult to see how families of diffeomorphisms of the higher-dimensional spacetime that depend on a number of arbitrary functions give rise to the Yang–Mills-type gauge symmetries of the lower-dimensional theory. Most of the lower-dimensional theories obtained by KK compactification also have global symmetries which, in the Einstein frame, only act on the matter fields. The origin of these global symmetries, formerly known as "hidden symmetries" and, more recently in the context of superstring/supergravity theories, as "dualities", is more mysterious.[2]

There are two main kinds of hidden symmetries:

1. Those which involve electric-magnetic-type duality transformations, which, typically, only leave invariant the equations of motion and can be considered non-perturbative.

2. The rest, which leave the action invariant.

The higher-dimensional origin of the former is unknown[3] and our results do not seem to provide new information about it. Thus, we are going to focus on the later, and, in particular, on symmetries which do not mix fields originating in different higher-dimensional fields.[4] For the sake of clarity, in this paper we will just consider pure $\hat{d}$-dimensional[5] gravity, described by the Einstein–Hilbert (EH) action, and its toroidal compactifications.

In most of the literature, the global symmetries of the theories obtained in these toroidal compactifications are understood as simple translations, rotations and rescalings (diffeomorphisms) in the internal directions. Closer inspection, though, shows that many of these diffeomorphisms are incompatible with the boundary conditions in the internal directions. Therefore, they are not diffeomorphisms of the toroidally compactified manifolds.

Let us start by considering the compactification on a circle, from $\hat{d}$ to $d = \hat{d} - 1$ dimensions of the EH action

$$S_{EH}[\hat{g}] \sim \int d^{\hat{d}}\hat{x} \sqrt{|\hat{g}|} \, \hat{R}(\hat{g}) \,. \tag{1.1}$$

---

[1]Many historical and more modern references on these theories can be found in Ref. [1].

[2]Explaining and making manifest all the global symmetries found in lower-dimensional string effective theories (supergravities) from the 10- and 11-dimensional point of view is the goal of double and exceptional field theory. For a recent review, see Ref. [2] and references therein.

[3]As we are going to explain, the standard explanation is not completely correct.

[4]In particular, this excludes T duality [3–10].

[5]We indicate with hats all higher-dimensional objects, except for the Killing vector $k$ and the vector $\epsilon$: $\{\hat{x}^{\hat{\mu}}\} = \{x^{\mu}, z\}$ etc. We use the notation and conventions of Ref. [11].

If we parametrize the compact direction with the periodic coordinate $z \sim z + 2\pi\ell$ adapted to the isometry $k = \partial_{\underline{z}}$, the $\hat{d}$-dimensional metric decomposes into a metric $g_{\mu\nu}$, a gauge field (the *KK vector*) $A = A_\mu dx^\mu$ and a scalar (the *KK scalar*) $k$

$$d\hat{s}^2 = \hat{g}_{\hat{\mu}\hat{\nu}} d\hat{x}^{\hat{\mu}} d\hat{x}^{\hat{\nu}} = ds^2 - k^2 (dz + A)^2 , \tag{1.2}$$

where

$$ds^2 = g_{\mu\nu} dx^\mu dx^\nu . \tag{1.3}$$

After integration over $z$, the $\hat{d}$-dimensional EH action can be rewritten, up to total derivatives, in the form

$$S[g, A, k] \sim \int d^d x \sqrt{|g|} \left\{ kR(g) - \tfrac{1}{4} k^3 F^2 \right\} , \tag{1.4}$$

where

$$F^2 = F_{\mu\nu} F^{\mu\nu} , \qquad F_{\mu\nu} = 2\partial_{[\mu} A_{\nu]} . \tag{1.5}$$

The factor $k$ in front of the Ricci scalar indicates that the metric $g_{\mu\nu}$ is not the Einstein-frame metric $g_{E\,\mu\nu}$. It is, however, related to it by

$$g_{\mu\nu} = k^{-2/(\hat{d}-3)} g_{E\,\mu\nu} = k^{-2/(d-2)} g_{E\,\mu\nu} , \tag{1.6}$$

and, if we rewrite the metric in the Einstein-frame, it takes the form

$$S[g_E, A, k] \sim \int d^d x \sqrt{|g_E|} \left\{ R(g_E) + \frac{d-1}{d-2} k^{-2} (\partial k)^2 - \tfrac{1}{4} k^{2\frac{d-1}{d-2}} F^2 \right\} . \tag{1.7}$$

This action is invariant under global rescalings of $k$ and $A$

$$\delta_\alpha A = \alpha A , \qquad \delta_\alpha k = -\alpha \frac{d-2}{d-1} k , \tag{1.8}$$

which do not act on the Einstein-frame metric.

As discussed, for instance, in Ref. [11], these transformations may be seen as the combination of a global rescaling of the coordinate $z$ and a global rescaling of the $\hat{d}$-dimensional metric

$$\delta_\alpha \hat{g}_{\hat{\mu}\hat{\nu}} = -2\alpha \delta^{\underline{z}}_{(\hat{\mu}} \hat{g}_{\hat{\nu})\underline{z}} + \frac{2\alpha}{\hat{d}-2} \hat{g}_{\hat{\mu}\hat{\nu}} , \tag{1.9}$$

where $\alpha$ is some constant, infinitesimal, parameter.

The rescaling of the coordinate $z$ is usually seen as a diffeomorphism generated by the vector

$$\hat{\eta} \equiv z\partial_{\underline{z}} , \tag{1.10}$$

so the first term in the transformation Eq. (1.9) is just

$$-\alpha \pounds_{\hat{\eta}} \hat{g}_{\hat{\mu}\hat{\nu}}\,, \tag{1.11}$$

where $\pounds_{\hat{\eta}}$ stands for the Lie derivative with respect to the vector field $\hat{\eta}$.

Such a rescaling, however, changes the periodicity of $z$ when it is periodic and cannot be consistently used in this setting. Another way of seeing this problem is to observe that the vector field $\hat{\eta}$ is not well defined when $z$ is periodic: it is not single valued. Furthermore, observe that, according to this interpretation, the complete transformation Eq. (1.9) would combine a symmetry of the action up to total derivatives (a diffeomorphism) with a transformation which is not a symmetry of the action (a global rescaling of the metric). Such a combination should not be a symmetry of the $\hat{d}$-dimensional theory!

As we are going to show, there is another, consistent, way of interpreting this transformation in which the first term in Eq. (1.9) corresponds to a transformation which is not a diffeomorphism nor a symmetry of the action, because it rescales it. This rescaling can be compensated with a global rescaling of the metric, resulting in a global symmetry of the $\hat{d}$-dimensional action which is inherited by the $d$-dimensional one and which corresponds, precisely, to the global symmetry in Eq. (1.8).

In toroidal compactifications, when there is more than one compact direction, the lower-dimensional theory is usually invariant under rescalings and also under rotations of the fields. They are usually understood as originating in rotations of the internal manifold. However, the putative $\hat{d}$-dimensional vector fields that would generated those rotations also have components proportional to compact coordinates and they are not single valued. Again, the rotations of the lower-dimensional fields are not diffeomorphisms in higher dimensions.

As we are going to see, the transformations which is not diffeomorphisms and which are associated to the rescalings and rotations of the compact coordinates are the analog of the higher-form symmetries of theories of $(p+1)$-form fields [12].[6] They are present in spaces of non-trivial topology. The symmetries that we have found also require an isometry, but they should be considered the simplest example of this kind of symmetries and an invitation to explore other possibilities.

This paper is organized as follows: in Section 2 we review higher-form symmetries in theories of $(p+1)$-form fields from the purely classical point of view in order to clarify in which sense the symmetries of the EH action that we are going to find in Section 3 are similar. In Section 4 we show how some of these symmetries give rise to the hidden symmetries that arise in toroidal KK compactifications. We discuss our results in Section 5 indicating possible extensions and future lines of research.

---

[6]For pedagogical reviews with many references, see, for instance Refs. [13, 14].

## 2 Higher-form symmetries

In this section we are going to review the so-called higher-form symmetries of $(p+1)$-form fields $A$ from a purely classical point of view. We will use a language close to the one we will employ in the gravity case which is our main interest and we will consider the simplest setting in which there is only one field of this kind in the theory and there are no Chern–Simons-like terms neither in the action nor in the field strength, which is, just, the $(p+2)$-form $F = dA$. It is convenient to use the language of differential forms in which the manifestly gauge-invariant action in $d$ dimensions takes the form

$$S[A] = \int_{\mathcal{M}} \frac{(-1)^{(p+1)d}}{2} F \wedge \star F \,. \tag{2.1}$$

Under an arbitrary variation of $A$

$$\delta S[A] = \int_{\mathcal{M}} \{\mathbf{E}_A \wedge \delta A + d\mathbf{\Theta}(A, \delta A)\} \,, \tag{2.2}$$

where

$$\mathbf{E}_A = -d \star F \,, \tag{2.3a}$$

$$\mathbf{\Theta}(A, \delta A) = \star F \wedge \delta A \,, \tag{2.3b}$$

are, respectively, the equation of motion of $A$ and the pre-symplectic potential.

The field strength $F$ is invariant under the transformations $\delta A$ which are closed

$$\delta F = d\delta A = 0 \,. \tag{2.4}$$

In a generic manifold $\mathcal{M}$, we can only count on the variations $\delta$ which are exact, *i.e.*

$$\delta_\chi A = d\chi \,, \tag{2.5}$$

where $\chi$ is an arbitrary $p$-form field. These are the standard gauge transformations of a $(p+1)$-form field. However, if $\mathcal{M}$ has some prescribed topology, there may be closed $(p+1)$-forms $\delta A$ which are not exact. In some cases (typically, in compact manifolds $\mathcal{M}$) we can find a basis of these forms $\{h_i^{(p+1)}\}$, and, then, both $F$ and the theory will be exactly invariant under linear combinations of them with arbitrary, constant, coefficients $a^i$:

$$\delta_a A = a^i h_i^{(p+1)} \equiv a^i \delta_i A \,. \tag{2.6}$$

These are global symmetries of the action and we must study them separately from the gauge ones.

Let us consider them first. The exact invariance of the action for any $\mathcal{M}$ in the same topology class implies the relation

$$d\Theta(A, \delta_a A) = -\mathbf{E}_A \wedge \delta_a A,\tag{2.7}$$

and for each independent global parameters $a^i$, we derive from it a relation of the form

$$d\mathbf{J}_i \doteq 0,\tag{2.8a}$$

$$\mathbf{J}_i = \Theta(A, \delta_i A) = \star F \wedge h_i^{(p+1)}.\tag{2.8b}$$

Each $(d-1)$-form $\mathbf{J}_i$ is the Hodge dual of the standard Noether current 1-form associated to the global invariance generated by the parameter $a^i$. The charges associated to these currents are computed by integrating over $(d-1)$-dimensional "volumes".

In the case of the gauge transformations Eq. (2.5), particularizing the generic variation Eq. (2.2) $\delta \to \delta_\chi$, integrating by parts and using the Noether identity $d\mathbf{E}_A = 0$, we find

$$\delta_\chi S[A] = \int_{\mathcal{M}} d\mathbf{J}[\chi],\tag{2.9a}$$

$$\mathbf{J}[\chi] \equiv \Theta(A, \delta_\chi A) + (-1)^{d-p-1} \mathbf{E}_A \wedge \chi.\tag{2.9b}$$

Since the action is exactly invariant under these transformations for any choice of $\mathcal{M}$, we conclude that

$$d\mathbf{J}[\chi] = 0,\tag{2.10}$$

off-shell. This happens because

$$\mathbf{J}[\chi] = d\mathbf{Q}[\chi],\tag{2.11a}$$

$$\mathbf{Q}[\chi] = (-1)^{d-p} \star F \wedge \chi.\tag{2.11b}$$

These results can be exploited to define charges that satisfy a Gauss law by integrating the $(d-2)$-form $\mathbf{Q}[\chi]$ over compact $(d-2)$-dimensional "surfaces" $\Sigma^{d-2}$

$$q[\chi] = \int_{\Sigma^{d-2}} \mathbf{Q}[\chi],\tag{2.12}$$

as follows [15]: observe that, by construction

$$d\mathbf{Q}[\chi] = \mathbf{J}[\chi] = \Theta(A, \delta_\chi A) + (-1)^{d-p-1} \mathbf{E}_A \wedge \chi.\tag{2.13}$$

The second term can be made to vanish by choosing a solution of the equations of motion $\mathbf{E}_A = 0$. Then, since the pre-symplectic potential is linear in $\delta A$, the first term can be made to vanish by choosing a *Killing* or *reducibility* parameter $\chi = \kappa$ such that

$$\delta_\kappa A = d\kappa = 0 \,. \tag{2.14}$$

Then, the charge $q[\kappa]$ satisfies a Gauss law because $d\mathbf{Q}[\kappa] = 0$. Again, for manifolds $\mathcal{M}$ with appropriate topology we can write the most general closed $\kappa$ in the form

$$\kappa = b^I h_I^{(p)} + de \,, \tag{2.15}$$

where the $b^i$ are constants and $\{h_I^{(p)}\}$ is a basis of closed but not exact $p$-forms. We can define an independent charge for each of them

$$q_I \equiv q[h_I^{(p)}] = (-1)^{d-p} \int_{\Sigma^{d-2}} \star F \wedge h_I^{(p)} \,, \tag{2.16}$$

but not for the exact part:

$$\int_{\Sigma^{d-2}} \star F \wedge de \sim \int_{\Sigma^{d-2}} d\left(\star F \wedge e\right) = \int_{\partial\Sigma^{d-2}} \star F \wedge e = 0 \,, \tag{2.17}$$

where we have used the equations of motion and the compactness of the integration surface. For this reason, the charges $q_I$ we have defined are insensitive to the ambiguity of the $h_I^{(p)}$, which are defined only up to the addition of a total derivative.

In the next section we are going to try to generalize this scheme in the context of gravity.

## 3 Higher-form symmetries in gravity

The EH action is invariant up to total derivatives under the following transformations of the metric field

$$\delta_\xi g_{\mu\nu} = -\pounds_\xi g_{\mu\nu} = -2\nabla_{(\mu}\xi_{\nu)} \,, \tag{3.1}$$

which are obviously associated to diffeomorphisms $\delta_\xi x^\mu = \xi^\mu$.

We would like to find more general transformations $\delta g_{\mu\nu}$ leaving the action invariant, at least if certain geometrical or topological conditions are met. In order to get some inspiration, let us consider the KK theory with one compact and isometric direction which we discussed in the introduction. In that case, both the transformations that become the gauge transformations of the KK vector in one dimension less and the transformations that rescale the KK vector and scalar in one dimension less are associated to diffeomorphisms generated by vector fields of the form

$$f(\hat{x})k \,, \qquad k = \partial_{\underline{z}} \,. \tag{3.2}$$

When the function $f(\hat{x})$ is an arbitrary function $\Lambda(x)$ of the $(\hat{d}-1)$-dimensional coordinates $x$, only, we get the gauge transformations of the KK vector field and when it is proportional to the isomeric coordinate $z$, it generates rescalings of the $z$ coordinate. We have argued that these rescalings are incompatible with the periodic boundary conditions of $z$, but let us ignore this fact for the moment and let us look at the transformations of the metric generated by vector fields of the above form. They act on the metric as

$$\delta_f \hat{g}_{\hat{\mu}\hat{\nu}} = -2\partial_{(\hat{\mu}} f k_{\hat{\nu})} \,, \tag{3.3}$$

because $k$ is a Killing vector that leaves invariant the metrics of the class considered in KK theory.

When $f = z$, the above transformation is well defined, even though we derived it from a vector field which is not. This fact suggests that we may try to search for transformations of the metric $g_{\mu\nu}$[7] of the form

$$\delta_\epsilon g_{\mu\nu} \equiv -2\epsilon_{(\mu} k_{\nu)} \,, \tag{3.4}$$

where $\epsilon = \epsilon^\mu \partial_\mu$ is some vector field and $k = k^\mu \partial_\mu$ is a Killing vector field of the metric, so that

$$\nabla_\mu k_\nu = \nabla_{[\mu} k_{\nu]} \,. \tag{3.5}$$

First of all, for the above transformation to be a symmetry in the class of metrics admitting $k$ as Killing vector, the following condition must be satisfied:

$$\pounds_k \left( \epsilon_{(\mu} k_{\nu)} \right) = 0 \,. \tag{3.6}$$

This condition is equivalent to

$$\pounds_k \hat{\epsilon} = \iota_k d\hat{\epsilon} + d\iota_k \hat{\epsilon} = 0, \tag{3.7}$$

which would be satisfied if (but not only if)

$$d\hat{\epsilon} = 0 \,, \tag{3.8a}$$

$$d\iota_k \hat{\epsilon} = 0 \,. \tag{3.8b}$$

We now want to see under which conditions these transformations leave the EH action invariant. We know that, when $\epsilon_\mu = \partial_\mu f$ or $\hat{\epsilon} = df$[8] for some well-defined $f$, the above transformation is associated to a diffeomorphism, much in the same way as the transformation $\delta A$ of a $(p+1)$-form field corresponds to a gauge transformation

---

[7]Now we consider a general metric in a general dimension and, henceforth, suppress the hats.

[8]In this context we denote with hats the 1-form $\hat{\epsilon} = \epsilon_\mu dx^\mu$ dual to the vector field $\epsilon = \epsilon^\mu \partial_\mu$.

when it is exact. Thus, we expect the transformation of EH action to be proportional to $d\hat{e}$ and, perhaps, to vanish when $\hat{e}$ is closed.

A straightforward calculation gives for the Levi-Civita connection

$$\delta_\epsilon \Gamma_{\mu\nu}{}^\rho = -g^{\rho\sigma} \left\{ k_\sigma \nabla_{(\mu} \epsilon_{\nu)} + k_\nu \nabla_{[\mu} \epsilon_{\sigma]} + k_\mu \nabla_{[\nu} \epsilon_{\sigma]} + 2\epsilon_{(\mu} \nabla_{\nu)} k_\sigma \right\}, \tag{3.9}$$

where we have used the Killing vector equation (KVE) Eq. (3.5) in the last step.

The variation of the Riemann curvature tensor is given by the Palatini identity

$$\delta_\epsilon R_{\alpha\mu\nu}{}^\rho = 2\nabla_{[\alpha|} \delta_\epsilon \Gamma_{|\mu]\nu}{}^\rho$$

$$= -g^{\rho\sigma} \left\{ \nabla_{[\alpha|} k_\sigma \nabla_{|\mu]} \epsilon_\nu + \nabla_{[\alpha|} k_\sigma \nabla_\nu \epsilon_{|\mu]} + \nabla_{[\alpha|} k_\nu \nabla_{|\mu]} \epsilon_\sigma - \nabla_{[\alpha|} k_\nu \nabla_\sigma \epsilon_{|\mu]} \right.$$

$$+ 2\nabla_\alpha k_\mu \nabla_{[\nu} \epsilon_{\sigma]} + 2\nabla_{[\alpha} \epsilon_{\mu]} \nabla_\nu k_\sigma + 2\nabla_{[\alpha|} \epsilon_\nu \nabla_{|\mu]} k_\sigma \tag{3.10}$$

$$+ k_\sigma \nabla_{[\alpha} \nabla_{\mu]} \epsilon_\nu + k_\sigma \nabla_{[\alpha|} \nabla_\nu \epsilon_{|\mu]} + k_\nu \nabla_{[\alpha} \nabla_{\mu]} \epsilon_\sigma - k_\nu \nabla_{[\alpha|} \nabla_\sigma \epsilon_{|\mu]}$$

$$+ 2k_{[\mu} \nabla_{\alpha]} \nabla_{[\nu} \epsilon_{\sigma]} + 2\epsilon_{[\mu} \nabla_{\alpha]} \nabla_\nu k_\sigma + 2\epsilon_\nu \nabla_{[\alpha} \nabla_{\mu]} k_\sigma \right\}.$$

In order to simplify the notation we define

$$P_{\epsilon\,\mu\nu} \equiv \nabla_{[\mu} \epsilon_{\nu]}, \qquad P_{k\,\mu\nu} \equiv \nabla_\mu k_\nu. \tag{3.11}$$

Using the Ricci identity and the integrability condition of the Killing vector

$$[\nabla_\mu, \nabla_\nu] \xi_\rho = -R_{\mu\nu\rho}{}^\sigma \xi_\sigma, \tag{3.12a}$$

$$\nabla_\mu \nabla_\nu k_\rho = -k^\sigma R_{\sigma\mu\nu\rho}, \tag{3.12b}$$

we can remove all the second derivatives and the transformation of the Riemann curvature tensor takes a much simpler form:

$$\delta_\epsilon R_{\alpha\mu\nu}{}^\rho = -g^{\rho\sigma} \left\{ -2P_{k\,[\alpha|\sigma} P_{\epsilon\,|\mu]\nu} + 2P_{k\,[\alpha|\nu} P_{\epsilon\,|\mu]\sigma} + 2P_{k\,\alpha\mu} P_{\epsilon\,\nu\sigma} + 2P_{k\,\nu\sigma} P_{\epsilon\,\alpha\mu} \right.$$

$$- k_\sigma R_{\alpha\mu\nu\lambda} \epsilon^\lambda - 2k_\sigma \nabla_{[\alpha|} P_{\epsilon\,|\mu]\nu} + 2k_\nu \nabla_{[\alpha} P_{\epsilon\,|\mu]\sigma} \tag{3.13}$$

$$+ 2k_{[\mu} \nabla_{\alpha]} P_{\epsilon\,\nu\sigma} - 2\epsilon_{[\mu} k^\lambda R_{\lambda|\alpha]\nu\sigma} - 2\epsilon_\nu k^\lambda R_{\lambda[\alpha\mu]\sigma} \right\}.$$

Replacing $\rho$ by $\mu$ and using

$$\nabla_{[\mu|}P_{\epsilon\,|\nu\rho]} = 0\,, \tag{3.14}$$

we get the transformation of the Ricci tensor

$$\delta_\epsilon R_{\alpha\nu} = -6P_{k\,(\alpha|\mu}P_{\epsilon\,|\nu)}{}^\mu + 2k_{(\alpha|}\nabla_\mu P_{\epsilon\,|\nu)}{}^\mu - 2k_\mu\nabla_{(\alpha|}P_{\epsilon\,|\nu)}{}^\mu - 2k^\lambda R_{\lambda(\alpha}\epsilon_{\nu)}\,. \tag{3.15}$$

Then

$$\delta_\epsilon\left(\sqrt{|g|}\,R\right) = \sqrt{|g|}\left[-\delta_\epsilon g_{\mu\nu}G^{\mu\nu} + g^{\mu\nu}\delta_\epsilon R_{\mu\nu}\right] \tag{3.16}$$

$$= -\epsilon^\mu k_\mu\sqrt{|g|}\,R + \sqrt{|g|}\left[-6\,P_k{}^{\mu\nu}P_{\epsilon\,\mu\nu} + 4k_\mu\nabla_\nu P_\epsilon{}^{\mu\nu}\right]\,.$$

We can eliminate the second term by demanding

$$P_{\epsilon\,\mu\nu} = 0\,, \quad \Rightarrow \quad d\hat{e} = 0\,, \tag{3.17}$$

which is the first of Eqs. (3.8). The first term may be eliminated by demanding $k^\mu\epsilon_\mu = \iota_k\hat{e} = 0$, which is a very strong condition. Instead, we observe that the result that we have obtained after demanding the closedness of $\hat{e}$, Eq. (3.17), is

$$\delta_\epsilon\left(\sqrt{|g|}\,R\right) = -\iota_k\hat{e}\left(\sqrt{|g|}\,R\right)\,, \tag{3.18}$$

which would be equivalent to a global rescaling of the action if

$$\iota_k\hat{e} = \text{constant}\,, \quad \Rightarrow \quad d\iota_k\hat{e} = 0\,, \tag{3.19}$$

which is the second of the consistency conditions Eqs. (3.8).

We can compensate this rescaling with another global rescaling

$$\delta_\alpha g_{\mu\nu} = \alpha g_{\mu\nu}\,, \quad \Rightarrow \quad \delta_\alpha\left(\sqrt{|g|}\,R\right) = \frac{(d-2)}{2}\alpha\left(\sqrt{|g|}\,R\right)\,, \tag{3.20}$$

choosing the parameter $\alpha = 2\iota_k\hat{e}/(d-2)$. Thus, combining these two transformations, we find that

$$\delta_\epsilon g_{\mu\nu} = -2\epsilon_{(\mu}k_{\nu)} + \frac{2}{(d-2)}\epsilon^\rho k_\rho g_{\mu\nu}\,, \tag{3.21}$$

leaves the EH action exactly invariant if the consistency conditions Eqs. (3.8) are met.

In the next section we are going to see in the KK context that these transformations include and extend the diffeomorphism invariance of the EH action and that, when

they are not diffeomorphisms, they generate the constant rescalings of the KK vector and scalar that leave invariant the $d = (\hat{d} - 1)$-dimensional action as well as the $\hat{d}$-dimensional one, as we have shown. Notice that, in the KK setting the dimensional parameter that occurs in the above formulae has to be replaced by $\hat{d}$.

## 3.1 Conserved charges

We have just shown that, under the conditions Eqs. (3.8), the transformations Eq. (3.21) leave the EH action exactly invariant

$$\delta_\epsilon S_{EH}[g] = 0 . \tag{3.22}$$

Associated to each of the closed, but not exact, 1-forms $\hat{\epsilon}$ satisfying also $d\iota_k \epsilon = 0$, $\{\epsilon_i\}$, there must be a conserved Noether current.

Under a general variation of the metric,

$$\delta S_{EH}[g] \sim \int_\mathcal{M} d^4x \left\{ \frac{\delta S}{\delta g_{\mu\nu}} \delta g_{\mu\nu} + \partial_\mu \Theta^\mu(g, \delta g) \right\} , \tag{3.23a}$$

$$\frac{\delta S}{\delta g_{\mu\nu}} = -\sqrt{|g|} G^{\mu\nu} , \tag{3.23b}$$

$$\Theta^\mu(g, \delta g) = \sqrt{|g|} \left[ g^{\mu\nu} \delta \Gamma_{\rho\nu}{}^\rho - g^{\rho\nu} \delta \Gamma_{\rho\nu}{}^\mu \right] . \tag{3.23c}$$

Then, for each $\epsilon_i$,

$$\partial_\mu \Theta^\mu(g, \delta_{\epsilon_i} g) = -\frac{\delta S}{\delta g_{\mu\nu}} \delta_{\epsilon_i} g_{\mu\nu} \doteq 0 . \tag{3.24}$$

Then, using Eq. (3.9) and the fact that global rescalings of the metric leave invariant the connection, the current $j_i{}^\mu$ is given by

$$j_i{}^\mu = \Theta^\mu(g, \delta_{\epsilon_i} g)$$

$$= \sqrt{|g|} \left\{ k^\mu \nabla_\rho \epsilon_i{}^\rho + 2\epsilon_i{}^\rho \nabla_\rho k^\mu \right\} , \tag{3.25}$$

where we have used the Killing equation satisfied by $k$ and the consistency conditions Eqs. (3.8).[9]

---

[9]For instance:

$$-k^\rho \nabla_\rho \epsilon_i{}^\mu = -k^\rho \nabla^\mu \epsilon_{i\rho} = -\nabla^\mu \left( k^\rho \epsilon_{i\rho} \right) + \epsilon_i{}^\rho \nabla^\mu k_\rho = 0 - \epsilon_i{}^\rho \nabla_\rho k^\mu = -\epsilon_i{}^\rho \nabla_\rho k^\mu . \tag{3.26}$$

We can check that, indeed, the above currents are conserved on-shell:

$$\nabla_\mu \left( j_i{}^\mu / \sqrt{|g|} \right) = -2\epsilon_i{}^\rho k^\lambda R_{\lambda\rho}$$

(3.27)

$$\doteq 0\,,$$

where we have used

$$k^\mu \nabla_\mu \nabla_\rho \epsilon_i{}^\rho = 0\,.$$

(3.28)

# 4 Higher-form symmetries in the Kaluza–Klein setting

Let us now consider the transformations that we have found in the preceding section in the KK setting in which one of the dimensions is compactified in a circle, parametrized by $z \in [0, 2\pi\ell]$, which is the coordinate adapted to an isometry, so there is always a Killing vector field $k = \partial_{\underline{z}}$. In this setting, the solution to the first of the consistency conditions Eqs. (3.8) is of the form[10]

$$\hat{\epsilon} = \beta dz + d\Lambda,$$

(4.1)

where $\beta$ is some constant. Then, the second equation reads

$$\beta + \partial_{\underline{z}}\Lambda = \alpha\,,$$

(4.2)

and is solved by

$$\Lambda = \Lambda(x) + (\alpha - \beta)z\,,$$

(4.3)

where $\partial_{\underline{z}}\Lambda(x) = 0$. The second term in $\Lambda$ actually gives a term of the same kind as the first. Furthermore, as we have discussed in the introduction, this second term is not single valued around the circle and, therefore, we must remove it setting $\alpha = \beta$. Thus,

$$\hat{\epsilon} = \alpha dz + d\Lambda(x)\,.$$

(4.4)

The exact part of $\hat{\epsilon}$ is associated to a well-defined vector field

$$\lambda \equiv \Lambda(x)\partial_{\underline{z}}\,,$$

(4.5)

[10]Observe that despite its local form, $dz$ is not an exact 1-form because $z$ is not a single-valued function. Strictly speaking, it is not a good coordinate, either: $S^1$ needs to be covered by, at least, two coordinate patches with coordinates $z_1$ and $z_2$ related by an additive constant in the overlap. In each patch, the 1-form we are denoting by $dz$ would be $dz_1$ and $dz_2$ and it would be closed, but there is not a single-valued function $f$ such that it is $df$ (exact). Most of the time it is simpler and sufficient to work with a single, periodically-identified, coordinate $z$ if one is careful.

which generates the gauge transformations of the KK vector field:

$$\delta_\lambda g_{\mu\nu} = \delta_\lambda \left( \hat{g}_{\mu\nu} - \hat{g}_{\underline{z}\mu} \hat{g}_{\underline{z}\nu} / \hat{g}_{\underline{z}\underline{z}} \right) = 0 \,, \tag{4.6a}$$

$$\delta_\lambda A_\mu = \delta_\lambda \left( \hat{g}_{\mu\underline{z}} / \hat{g}_{\underline{z}\underline{z}} \right) = -\partial_\mu \Lambda \,, \tag{4.6b}$$

$$\delta_\lambda k = \delta_\lambda |\hat{g}_{\underline{z}\underline{z}}|^{1/2} = 0 \,. \tag{4.6c}$$

The global part of $\hat{\epsilon}$ acts on the $\hat{d}$-dimensional metric as

$$\delta_\alpha \hat{g}_{\hat{\mu}\hat{\nu}} = -2\alpha \delta^{\underline{z}}_{(\hat{\mu}} \hat{g}_{\hat{\nu})\underline{z}} + \frac{2}{(\hat{d}-2)} \alpha \hat{g}_{\hat{\mu}\hat{\nu}} \,. \tag{4.7}$$

These transformations act in a non-trivial way over all the fields in the $d$-dimensional KK frame. In the Einstein frame, though, only the KK scalar and vector transform

$$\delta_\alpha g_{E\,\mu\nu} = \delta_\alpha \left( k^{2/(\hat{d}-3)} g_{\mu\nu} \right) = 0 \,,$$

$$\delta_\alpha A_\mu = \alpha A_\mu \,, \tag{4.8}$$

$$\delta_\alpha k = -\alpha \frac{(\hat{d}-3)}{(\hat{d}-2)} k \,,$$

and these transformations coincide precisely with those in Eq. (1.8) that leave the Einstein-frame action Eq. (1.7) invariant.

Thus, we have shown that this global symmetry of the compactified theory is related to a symmetry of the higher-dimensional EH action which is not a diffeomorphism. This symmetry has been constructed with the help of a global rescaling of the metric, but we are going to see that in more general toroidal compactifications we do not need this global rescaling and we have symmetries originating only in the global part of $\hat{\epsilon}$.

## 4.1   Toroidal compactifications

In toroidal compactifications there are $n$ compact, mutually commuting isometries generated by Killing vectors that can be expressed in adapted coordinates as

$$k_m = k_m{}^{\hat{\mu}} \partial_{\hat{\mu}} = \partial_{\underline{z}^m} \equiv \partial_m \,. \tag{4.9}$$

The isometric coordinates $z^m$ parametrize the $n$ circles and, for simplicity, we assume that all of them have the same period $z^m \sim z^m + 2\pi\ell$. The dimensional reduction of the $\hat{d}$-dimensional EH action can be performed in two steps. First, we decompose the $\hat{d}$-dimensional metric $\hat{g}_{\hat{\mu}\hat{\nu}}$ into $d = (\hat{d}-n)$-dimensional fields: a KK-frame metric $g_{\mu\nu}$, $n$ gauge fields (*KK vectors*) $A^m = A^m{}_\mu dx^\mu$ and $n(n+1)/2$ *KK scalar fields* described by a symmetric, positive-definite matrix $G_{mn}$

$$ds^2 = \hat{g}_{\hat{\mu}\hat{\nu}}d\hat{x}^{\hat{\mu}}d\hat{x}^{\hat{\nu}} = ds^2 - G_{mn}(dz^m + A^m)(dz^n + A^n)\,, \tag{4.10}$$

where

$$ds^2 = g_{\mu\nu}dx^\mu dx^\nu\,. \tag{4.11}$$

After integration over the $n$ isometric directions, we obtain the $d$-dimensional KK-frame action

$$S[g, A^m, G_{mn}] \sim \int d^d x \sqrt{|g|}\, K \left\{ R(g) - (\partial \ln K)^2 - \tfrac{1}{4}\partial_\mu G_{mn}\partial^\mu G^{mn} - \tfrac{1}{4}F^2 \right\}\,, \tag{4.12}$$

where

$$K^2 \equiv |\det(G_{mn})|\,, \qquad F^2 \equiv G_{mn}F^{m\,\mu\nu}F^n{}_{\mu\nu}\,, \qquad F^m{}_{\mu\nu} \equiv 2\partial_{[\mu}A^m{}_{\nu]}\,. \tag{4.13}$$

The second step consists in a rescaling of the KK-frame metric to the Einstein-frame metric. It is also convenient to rescale the matrix of scalars to obtain a unimodular matrix $\mathcal{M}$:

$$g_{\mu\nu} = K^{-\frac{2}{d-2}}g_{E\,\mu\nu}\,,$$

$$\tag{4.14}$$

$$G_{mn} \equiv K^{\frac{2}{n}}\mathcal{M}_{mn}\,.$$

The result is the following $d$-dimensional Einstein-frame action

$$S[g, A^m, K, \mathcal{M}_{mn}] \sim \int d^d x \sqrt{|g_E|} \left\{ R_E + \tfrac{1}{2}\left(\partial \ln K^{-2a}\right)^2 - \tfrac{1}{4}\partial_\mu \mathcal{M}_{mn}\partial^\mu \mathcal{M}^{mn} \right.$$

$$\tag{4.15}$$

$$\left. -\tfrac{1}{4}K^{(2a)^2}\mathcal{M}_{mn}F^{m\,\mu\nu}F^n{}_{\mu\nu} \right\}\,,$$

where we have defined the constant

$$a \equiv -\sqrt{\frac{(d-2+n)}{2n(d-2)}}\,. \tag{4.16}$$

This action is invariant under $\mathrm{SL}(n, \mathbb{R})$ transformations which only act on the KK vectors and $\mathcal{M}$

$$A'^m = S^m{}_n A^n\,, \qquad \mathcal{M}'_{mn} = S^{-1\,p}{}_m S^{-1\,q}{}_n \mathcal{M}_{pq}\,, \tag{4.17}$$

and which are only non-trivial for $n \geq 2$ and under global rescalings which only act on the KK scalar $K$ and the KK vectors $A^m$

$$K' = C^{-\frac{1}{2a^2}} K, \qquad A'^m = CA^m, \tag{4.18}$$

where $C$ is an arbitrary positive real constant. Together, they generate the $\mathrm{GL}(n, \mathbb{R})$ duality group of this theory which, in the literature, is customarily associated to diffeomorphisms of the internal space $\mathrm{T}^n$. However, just as in the $n = 1$ case, most of those transformations do not preserve the boundary conditions and are generated by vector fields which are not single valued.

In this case, we can consider a linear combination of transformations of the form Eq. (3.21) for each of the $n$ Killing vectors. We have to introduce $n$ closed 1-forms $\hat{\epsilon}^m$ in order to construct transformations $\delta_\epsilon g_{\mu\nu} \sim \epsilon^{(m)}{}_{(\mu|} k_{(m) |\nu)}$, (no sum over $m$ intended) but, once we have introduced them, we can obviously consider other pairings $\epsilon^m{}_{(\mu|} k_{n |\nu)}$, $m \neq n$ and, therefore, we are led to consider the most general possibility

$$\delta_\epsilon \hat{g}_{\hat{\mu}\hat{\nu}} = -2T^m{}_n \epsilon^n{}_{(\hat{\mu}|} k_{m |\hat{\nu})}, \tag{4.19}$$

where $T^m{}_n$ is a matrix of constant, infinitesimal, parameters. Each of the terms in the linear combination of the right-hand side must satisfy the condition

$$\epsilon^n{}^{\hat{\rho}} k_{m \hat{\rho}} = \text{constant}, \tag{4.20}$$

and, furthermore, it needs to be compensated by a global rescaling in order to generate a symmetry of the EH action. Thus, we must consider the transformations Eq. (4.19) supplemented by a global rescaling, if necessary.

In this setting, we can always choose the closed 1-forms $\epsilon^m$ so that

$$\epsilon^n{}^{\hat{\rho}} k_{m \hat{\rho}} = \delta^n{}_m, \quad \Rightarrow \quad \hat{\epsilon}^m = dz^m + d\Lambda^m(x). \tag{4.21}$$

The exact part does not need to be supplemented by global rescalings. It generates gauge transformations of the KK vectors

$$\delta_\Lambda A^m = d\Lambda'^m, \qquad \Lambda'^m = T^m{}_n \Lambda^n. \tag{4.22}$$

In what follows, we will only consider the non-exact part. Taking into account the necessary global rescalings, the transformations take the form

$$\delta_\epsilon \hat{g}_{\hat{\mu}\hat{\nu}} = -2T^p{}_q \epsilon^q{}_{(\hat{\mu}|} k_{p |\hat{\nu})} + \frac{2}{(\hat{d} - 2)} T^p{}_p \hat{g}_{\hat{\mu}\hat{\nu}}. \tag{4.23}$$

Since only the trace of $T^m{}_n$ needs to be compensated by the global rescalings, we decompose it into its traceless and trace parts:

$$T^m{}_n = T^m{}_n - \frac{1}{n} \delta^m{}_n T^p{}_p + \frac{1}{n} \delta^m{}_n T^p{}_p \equiv R^m{}_n + T \delta^m{}_n, \tag{4.24}$$

and we end up with

$$\delta_\epsilon \hat{g}_{\hat{\mu}\hat{\nu}} = -2R^p{}_q \epsilon^q{}_{(\hat{\mu}|} k_{p\,|\hat{\nu})} - 2T\left[\epsilon^p{}_{(\hat{\mu}|} k_{p\,|\hat{\nu})} - \frac{n}{(\hat{d}-2)}\hat{g}_{\hat{\mu}\hat{\nu}}\right]. \tag{4.25}$$

We can define two independent sets of transformations:

$$\delta_R \hat{g}_{\hat{\mu}\hat{\nu}} \equiv -2R^p{}_q \epsilon^q{}_{(\hat{\mu}|} k_{p\,|\hat{\nu})}, \qquad R^p{}_p = 0, \tag{4.26a}$$

$$\delta_T \hat{g}_{\hat{\mu}\hat{\nu}} \equiv -2T\left[\epsilon^p{}_{(\hat{\mu}|} k_{p\,|\hat{\nu})} - \frac{n}{(\hat{d}-2)}\hat{g}_{\hat{\mu}\hat{\nu}}\right]. \tag{4.26b}$$

Taking into account that

$$\epsilon^p{}_{(\hat{\mu}|} k_{q\,|\hat{\nu})} = \delta^p{}_{(\hat{\mu}|} \hat{g}_{q\,|\hat{\nu})}, \tag{4.27}$$

the effect of the $\delta_T$ transformations on the $d$-dimensional Einstein-frame fields is

$$\delta_T K = -\frac{n(\hat{d}-2-n)T}{(\hat{d}-2)}K = -\frac{n(d-2)T}{(d-2+n)}K = -\frac{T}{2a^2}K,$$

$$\delta_T A^m{}_\mu = TA^m{}_\mu, \tag{4.28}$$

and the Einstein metric and $\mathcal{M}_{mn}$ are invariant. These transformations are the infinitesimal version of those in Eq. (4.18), which leave the Einstein-frame action invariant.

The $\delta_R$ transformations act on the $d$-dimensional fields as

$$\delta_R \mathcal{M}_{mn} = -2R^p{}_{(m|}\mathcal{M}_{p\,|n)},$$

$$\delta_R A^m{}_\mu = R^m{}_n A^n{}_\mu, \tag{4.29}$$

leaving the rest invariant.

## 4.2 Conserved charges

Let us compute the $\hat{d}$-dimensional Noether current of the $T$ and $R$ symmetries. Using the general expression Eq. (3.25), we find

$$j^m{}_n{}^{\hat{\mu}}/\sqrt{|\hat{g}|} = k_n{}^{\hat{\mu}}\hat{\nabla}_{\hat{\rho}}\epsilon^{m\,\hat{\rho}} + 2\epsilon^{m\,\hat{\rho}}\hat{\nabla}_{\hat{\rho}}k_n{}^{\hat{\mu}}$$

$$= -\delta_n{}^{\hat{\mu}}\frac{1}{\sqrt{|g|}\,K}\partial_\rho\left(\sqrt{|g|}\,K\,A^{m\rho}\right) - 2A^{m\rho}\hat{\Gamma}_{\rho n}{}^{\hat{\mu}} \tag{4.30}$$

$$- 2\left(G^{mp} - A^m{}_\rho A^{p\,\rho}\right)\hat{\Gamma}_{pn}{}^{\hat{\mu}},$$

where we have decomposed the $\hat{d}$-dimensional fields in terms of the $d$-dimensional ones. The $d$-dimensional components are

$$j^m{}_n{}^\mu / \sqrt{|\hat{g}|} = K^{\frac{2}{d-2}} \left[ K^{(2a)^2} \mathcal{M}_{np} F^p{}_\rho{}^\mu A^{m\rho} - \mathcal{M}^{mp} \partial^\mu \mathcal{M}_{pn} - \frac{2}{n} K^{-1} \partial^\mu K \delta^m{}_n \right] . \tag{4.31}$$

where we have expressed the current in the Einstein-frame metric.

The trace part is, (again in the Einstein frame)

$$j^m{}_m{}^\mu / \sqrt{|g|} \sim 2K^{-1} \partial^\mu K + K^{(2a)^2} \mathcal{M}_{mn} F^m{}^\mu{}_\rho A^{n\rho} , \tag{4.32}$$

and it is easy to check that its divergence is proportional a combination of the $d$-dimensional equations of motion of $K$ and $A^m$ and vanishes on-shell.

The traceless part is

$$R^n{}_m j^m{}_n{}^\mu / \sqrt{|g|} \sim R^n{}_m \left[ \mathcal{M}^{mp} \partial^\mu \mathcal{M}_{pn} + K^{(2a)^2} \mathcal{M}_{np} F^{p\,\mu}{}_\rho A^{m\rho} \right] , \tag{4.33}$$

and its divergence is a combination of the equations of motion of $A^m$ and

$$R^p{}_m \mathcal{M}_{np} \frac{\delta S}{\delta \mathcal{M}_{mn}} , \tag{4.34}$$

which, therefore, also vanishes on-shell.

Both currents coincide with the Noether currents of the $T$ and $R$ symmetries of the $d$-dimensional action, as expected.

# 5 Discussion

In this paper we have shown that the Einstein–Hilbert action is invariant under transformations of the metric which are not diffeomorphisms. In the KK stting that we have chosen as an example, these transformations are equivalent to diffeomorphisms which are not globally well defined,[11] and they give well-known symmetries of the compactified theory. Nevertheless, these symmetries provide a higher-dimensional explanation for them which, strictly speaking, was not available in the literature. On the other hand, we believe that the symmetries that we have found are just the simplest in their class and that, in more general settings, the Einstein–Hilbert action will certainly admit more general non-diffeomorphic symmetries.

There is another interesting aspect of the relation between global duality symmetries in compactified theories and non-diffeomorphic symmetries in higher dimensions. It is believed that Quantum Gravity theories should not have any global symmetries [16–19], but if these symmetries were just a global subgroup of a gauge group, it would be very difficult to argue that only that particular subset should be broken.

---

[11]This is always going to be the case, since all closed 1-forms are locally exact.

Our results imply (at least in the simple examples tat we have explored here) that those global duality symmetries are not a subgroup of the group of diffeomorphisms and, therefore, they can be broken while preseving the integrity of the group of diffeomorphisms.

It should also be clear that the coupling to matter may modify or enhance the set of non-diffeomorphic symmetries, mixing now different higher-dimensional fields as it happens in T duality.

Finally, we know that, when we compactify a theory, the existence of global symmetries in a theory allows for *generalized dimensional reduction* ansatzs in which one performs a global symmetry transformation of the matter fields with a parameter which is linear in one of the compact coordinates. These ansatzs lead to gauge/massive theories in lower dimensions. A good example is provided by the generalized dimensional reduction of type IIB supergravity from 10 to 9 dimensions exploting the full $SL(2, \mathbb{R})$ global symmetry of the theory performed in Ref. [9]. These generalized dimensional reductions can also be associated to the introduction of non-dynamical branes in the background [21,9]. There is another kind of generalized (*Scherk–Schwarz*) dimensional reduction ansatz, proposed in Ref. [20], which may be related to the kind of global symmetries acting on the metric that we have been discussing in this paper. In future work we would like to explore the possible connection between the Scherk–Schwarz ansatz and the global, non-diffeomorphic symmetries identified in this paper.

# Acknowledgments

TO would like to thank J.J. Fernández-Melgarejo and A. Rosabal for interesting conversations and for their hospitality at the U. of Murcia. This work has been supported in part by the MCIU, AEI, FEDER (UE) grant PID2021-125700NB-C21 ("Gravity, Supergravity and Superstrings" (GRASS)) and IFT Centro de Excelencia Severo Ochoa CEX2020-001007-S. The work of CG-F was supported by the MU grant FPU21/02222. The work of MZ was supported by the fellowship LCF/BQ/DI20/11780035 from "la Caixa" Foundation (ID 100010434). TO wishes to thank M.M. Fernández for her permanent support.

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
