# Peer review of "Gravitational higher-form symmetries and the origin of hidden symmetries in Kaluza-Klein compactifications"

_SciPost Physics Core_

## Round 1 · Referee Report · Anonymous (Referee 1) · 2024-9-6

Strengths

1) Good presentation 2) Correct results 3) Important observations 4) Wide scope of applications

Weaknesses

None.

Report

The paper shows that in addition to diffeomorphisms, the Einstein-Hilbert action is invariant under a previously unknown global symmetry (3.21).
It is not a standard symmetry as it holds for a subset of configurations with isometries and non-trivial topology, but since operationaly no specific details on the backgrounds are required, for all practical purposes it works as an off-shell symmetry prinicple at the level of the action. It is generated by a 1-form whose contraction with the killing vector is constant, and is closed but no exact (3.8), and for this reason the authors refer to it as a higher-form gravitational symmetry. Afetr a Kaluza-Klein reduction, it descends to a known symmetry in the lower diemnsional action.

A new invariance of the Einstein-Hilbert action is definitely an important result. The paper is clearly written and well organized, the computations are certainly correct, the results are intriguing and thought provoking and the scope of applications apears to be wide. I recommend the paper for publication as it stands.

As a small suggestion for future work, it could be interesting to explore the closure relations with diffeomorphisms, to understand the underlying algebraic structure, and see if they form a complete set of symmetries.

Requested changes

Is the second equality of Eq. (2.8b) a typo?

Recommendation

Publish (meets expectations and criteria for this Journal)

  • validity: high
  • significance: high
  • originality: high
  • clarity: high
  • formatting: excellent
  • grammar: perfect

Author:  Tomás Ortín  on 2024-09-12  [id 4776]

(in reply to Report 1 on 2024-09-06)
Category:
answer to question

The second equality of Eq. (2.8b) is correct. It is the exterior product of a (p+1)-form and a (d-p-2)-form, which gives a (d-1)-form, the dual of a standard Noether current (we always use the dual forms, that can be directly integrated over (d-1)-dimensional hypersurfaces). On the other hand, this equation follows directly from (2.3b) and (2.6).

---

## Round 1 · Referee Report · Anonymous (Referee 2) · 2024-9-10

Strengths

1- The paper makes an attempt to clarify and render more exact some often sloppily used notion of the higher-dimensional origin of certain global symmetries.

Weaknesses

1- According to the discussion in the introduction (after (1.11)): a main result is the new "global symmetry of the $\hat{d}$-dimensional action" as stated in the introduction after (1.11) which is then "inherited by the $d$-dimensional one”. However, the former one is only derived under the assumption of the existence of a Killing vector. Effectively this is already in the framework of a $d$-dimensional action. Constructing a higher-dimensional origin of a symmetry action using a Killing vector is essentially nothing but a rewriting of the lower-dimensional structure. It is not clear what can be learned from this.

2- It is not clear to what extent these results can improve the understanding of either, the higher- or the lower-dimensional theory. Nor if they allow for any further applications.

Report

This short paper aims at setting up relations between certain hidden symmetries and a proposed gravitational analogue of higher-form symmetries. However, the "new invariances" of the action only hold in presence of Killing vector fields. It should be justified why this structure is more than a mere rewriting of the well-known lower-dimensional symmetries.

Requested changes

various imprecise statements and notation:

1- The use of ‘hidden symmetries’ in title and abstract as well as their discussion in the introduction is rather misleading. The split into two kinds, labelled 1. and 2. in the introduction, mixes up the notion of ‘hidden symmetry’ in the commonly used sense (no obvious higher-dimensional origin) with the unrelated issue of invariance of an action vs equations of motion (which only arises in even dimensions). By this definition, the exceptional group E7 in four-dimensional maximal supergravity would fall into class 1 but the exceptional group E6 in five-dimensional maximal supergravity would fall into class 2. Yet, there is no obvious difference regarding the status of the higher-dimensional origin of these groups. The paper then restricts to pure gravity theories whose symmetries are rarely described as ‘hidden’ (in compactifications above three dimensions).

2- footnote 3: is the higher-dimensional origin of the former “unknown” as stated in the main text, or is there a “standard explanation” which is “not correct”, as stated in the footnote?

3- clarify the notion of $z$ vs ${\underline{z}}$

4- notation $\hat{\epsilon}$ is only introduced in footnote 8, but already used in equation (3.7)

5- equations (3.8a) (3.8b) are sufficient but not necessary for (3.7), why are these stronger equations introduced?

6- text before (3.9) “Thus, we expect the transformation of EH action to be proportional to $d\hat\epsilon$ and, perhaps, to vanish when $\hat\epsilon$ is closed.” Why ‘perhaps’? Would the transformation not necessarily vanish if it is proportional to $d\hat\epsilon=0$?

7- in (3.16), why do the terms have to vanish separately?

8- in (3.19) why are the statements not equivalent?

9- what is meant by “exactly invariant” before (3.22)?

10- what is $\alpha$ in (4.2)? is it defined by this equation? There is no $\alpha$ in (3.8) or the ansatz (4.1).

11- most importantly, the above weaknesses should be addressed: It should be justified why the new structure is more than a mere rewriting (in terms of a Killing vector) of the well-known lower-dimensional symmetries.

Recommendation

Ask for major revision

---

## Editorial Decision

resubmitted